# CROSS-LINGUAL DATA SCALING FOR LARGE LANGUAGE MODELS

## ABSTRACT

Large language models (LLMs) achieve consistent performance gains through data scaling, yet low-resource languages remain limited by small and stagnant dataset sizes. To address this limitation, we introduce cross-lingual data scaling, where performance in low-resource languages scales with the dataset size of high-resource languages. We systematically investigate two potential approaches: (i) transforming high-resource language data into synthetic data for low-resource languages via translation or code-switching, and (ii) transferring the learned knowledge from high-resource languages to low-resource languages by adjusting language order and proportion during pretraining. Experiments on English and Chinese show that data transformation fails to sustain cross-lingual data scaling, whereas knowledge transfer enables low-resource language performance to scale with the growth of high-resource language data. Building on these findings, we propose ScaleX, a two-stage pretraining framework designed for effective cross-lingual data scaling. In the first stage, LLMs are pretrained on high-resource language data under a constant learning rate schedule; in the second stage, training continues on a mixture of high- and low-resource languages under a cosine learning rate schedule. ScaleX outperforms existing approaches with progressively larger margins as high-resource data scales up, and further generalizes to both multilingual and large-scale bilingual pretraining. Our analysis also reveals that learning rate scheduling and shared tokens across languages are critical to sustaining performance scaling in low-resource languages.

## 1 INTRODUCTION

Large language models (LLMs) have achieved significant advances through data scaling (Kaplan et al., 2020; Hoffmann et al., 2022), as performance improves consistently with larger training datasets. However, for many low-resource languages, the benefits of data scaling are constrained by small and stagnant dataset sizes (Yu et al., 2022; Ranathunga & de Silva, 2022). This constraint raises a fundamental challenge: *how can LLMs sustain performance scaling in low-resource languages when their training data is limited?*

In multilingual pretraining, data scaling is predominantly driven by high-resource language data (Conneau et al., 2020; Xue et al., 2021; Li et al., 2025b). This dominance motivates us to explore **cross-lingual data scaling**, where LLM performance in low-resource languages scales with the dataset size of high-resource languages. Unlike conventional data scaling, cross-lingual data scaling inherently entails a mismatch between training and evaluation languages. To address this mismatch, it is necessary to adapt either the training or the evaluation language to the other. This adaptation can be approached in two ways: (i) transforming high-resource language data into synthetic training data for low-resource languages; and (ii) transferring the learned knowledge from high-resource languages to low-resource languages.

In this paper, we take English and Chinese as illustrative examples of high- and low-resource languages, and pretrain LLMs from scratch to explore the feasibility of cross-lingual data scaling. To explore the potential of data transformation, we examine two widely used approaches for generating synthetic data for Chinese: translation (Joshi et al., 2025; Wang et al., 2025a), which converts English data into Chinese using open-source LLMs, and code-switching (Yoo et al., 2025; Wang et al., 2025b), which mixes Chinese tokens or sentences into English samples. Experimental results

show that while scaling such synthetic data mitigates performance degradation, its limited quality prevents performance from scaling effectively.

To explore the potential of knowledge transfer, we analyze the effect of scaling raw English data on Chinese performance and observe that Chinese performance consistently declines as English data increases. This decline occurs because, under uniform mixing of bilingual data, the proportion of limited Chinese data decreases as the English data increases. To overcome this negative transfer, we adjust the language ordering (Zheng et al., 2024; Bari et al., 2025) and the proportion (Wei et al., 2023; Yue et al., 2025) so that the limited Chinese data are concentrated in the later stage of pretraining with a higher proportion. Experimental results show that in a two-stage pretraining setup, expanding English data in the first stage enables Chinese performance to scale with English data growth, while increasing the proportion of Chinese data in the second stage leads to further performance improvements.

Building on these findings, we propose ScaleX, a pretraining framework that enables effective cross-lingual data scaling. ScaleX adopts a two-stage paradigm. In the first stage, LLMs are pretrained on high-resource language data under a constant learning rate schedule, which lays the foundation for scalable transfer. In the second stage, training continues on a mixture of high- and low-resource languages under a cosine learning rate schedule, further enhancing low-resource language performance. Beyond this training setup, our further analysis reveals two key factors underlying ScaleX's effectiveness: the learning rate schedule, which ensures effective training on low-resource language data in the second stage, and the presence of shared tokens across languages, such as code-switching tokens, punctuation, and digits, which enable the scalability of low-resource performance as high-resource data increases.

Our main contributions are as follows:

- We introduce cross-lingual data scaling and conduct a comprehensive analysis of potential approaches.
- We propose ScaleX, a scalable cross-lingual pretraining framework that achieves sustained gains for low-resource languages, outperforming existing approaches by progressively larger margins as high-resource language data scales up.
- We validate ScaleX in both multilingual pretraining and large-scale bilingual pretraining, demonstrating its practicality and generalization.
- We reveal the underlying mechanisms by which ScaleX sustains performance scaling in low-resource languages, providing clear guidance for future research on low-resource language modeling.

## 2 RELATED WORK

**Cross-Lingual Transfer**  To address the data imbalance between high- and low-resource languages in LLM pretraining, previous studies on cross-lingual transfer can be broadly categorized into three directions: (i) Data strategies primarily address the imbalance by upsampling low-resource language data (Lin et al., 2024; Penedo et al., 2025; He et al., 2025) or generating synthetic data via translation (Joshi et al., 2025; Wang et al., 2025a) or code-switching (Yoo et al., 2025; Wang et al., 2025b). (ii) Training objectives focus on reweighting the loss function, placing greater weight on losses from low-resource languages (Fan et al., 2025). (iii) Training strategies adjust the scheduling of multilingual data during pretraining, for example, through changes in language order (Zheng et al., 2024; Bari et al., 2025) or adjustments in the proportions of high- and low-resource language data (Wei et al., 2023; Yue et al., 2025). In contrast to prior work that primarily focuses on the magnitude of performance improvements in low-resource languages, our focus is on whether such improvements scale with the increasing amount of high-resource language data.

**Strategies for Data Exhaustion**  The data exhaustion challenge (Villalobos et al., 2022) in LLM scaling has been primarily mitigated through two approaches: data repetition and data filtering. In terms of data repetition, Muennighoff et al. (2023) demonstrate that repeating training data across multiple epochs enables LLM scaling to continue under data-constrained conditions. Xue et al. (2023) highlights the token-crisis problem and proposes regularization methods to alleviate overfitting and performance decline when training data is scarce and reused repeatedly. Tirumala et al.

(2023) introduces document de-duplication and diversification to enhance the effectiveness of pre-training on repeated data. In terms of data filtering, Muennighoff et al. (2023); Nguyen et al. (2025) propose enhanced strategies to better preserve useful data for LLM pretraining. However, filtering methods provide only a marginal increase in pretraining data, and this often comes at the cost of adding lower-quality content. Moreover, repeated use of data leads to diminishing returns and can even degrade performance (Hernandez et al., 2022; Muennighoff et al., 2023). In contrast to these limitations, our work introduces cross-lingual data scaling, which enables performance in low-resource languages to scale with the dataset size of high-resource languages.

# 3 PRELIMINARY STUDY

In this section, we explore the feasibility of cross-lingual data scaling from two perspectives: data transformation and knowledge transfer.

## 3.1 EXPERIMENTAL SETUP

**Dataset** LLMs are pretrained on Chinese and English corpora drawn from diverse publicly available sources, including webpages, code, Wikipedia, papers, books, question answering datasets, examinations, mathematics, knowledge bases, translations, and other open sources. To simulate cross-lingual data scaling, we fix the Chinese dataset size at 50B tokens and vary the English dataset size across 150B, 250B, and 450B tokens. This setup results in total dataset sizes of 200B, 300B, and 500B tokens.

**Model and Optimization** Experiments are conducted using LLaMA architecture (Touvron et al., 2023a;b) with 1.3B parameters, randomly initialized and pretrained from scratch. All LLMs are trained with the Megatron-LM framework (Shoeybi et al., 2019), which provides efficient parallelism and scalability for large-scale pretraining. We adopt a cosine learning rate schedule, where the learning rate follows a cosine decay from $3 \times 10^{-4}$ to $3 \times 10^{-5}$. Each LLM is optimized with a maximum input sequence length of 4,096 tokens and a global batch size of 1,024.

**Evaluation** We evaluate LLM performance using both perplexity (PPL) and downstream task accuracy. PPL is evaluated in both English and Chinese. For Chinese downstream evaluation, we employ widely used benchmarks, including CMMLU (Li et al., 2024), C-EVAL (Huang et al., 2023), AGIEval (Zhong et al., 2024), and Math23K (Wang et al., 2017). For English evaluation, we employ GSM8K (Cobbe et al., 2021), BBH (Suzgun et al., 2023), MMLU (Hendrycks et al., 2021), and HumanEval (Chen et al., 2021).

## 3.2 COMPARISON APPROACHES

To explore cross-lingual data scaling, we adopt several approaches and assess their behavior as the amount of high-resource language data increases. The approaches are described below:

- **Baseline** directly scales up the pretraining dataset of high-resource languages. As the dataset size of low-resource languages remains fixed, the proportion of low-resource language data diminishes as high-resource language data grows.

- **Translation** converts high-resource language samples into low-resource language samples using Qwen2.5-7B-Instruct (Yang et al., 2024). The translated data is incorporated into the pretraining dataset, replacing an equal amount of high-resource samples to maintain a constant total dataset size. Further details are provided in Appendix A.1.

- **Code-switching** injects low-resource language tokens or sentences into high-resource language samples, producing mixed-language data for low-resource language training. Details are given in Appendix A.2.

- **Order Adjustment** adopts a two-stage pretraining paradigm in which the model is first trained on high-resource language data, followed by training on a balanced mixture of high- and low-resource language data. Since the amount of low-resource language data is fixed, the number of training steps in the second stage is determined by its availability

- **Data Repetition** duplicates low-resource language data and replaces an equal amount of high-resource data with the repeated samples to maintain language balance.

- **Loss Weighting** increases the loss weight of low-resource languages in proportion to the size ratio between high- and low-resource datasets, which has been shown to yield effects similar to data repetition (Li et al., 2025a).

## 3.3 DATA TRANSFORMATION FOR CROSS-LINGUAL DATA SCALING

**Translation.** We first investigate translation, a widely used approach for augmenting low-resource language data. English samples are translated into Chinese using an open-source LLM, and the resulting synthetic data replaces the original English samples in the pretraining dataset to keep the total size constant. As shown in Figure 1, translation alleviates performance degradation compared to baseline, but Chinese performance still declines as English data increases. This suggests that translation alone is insufficient to achieve cross-lingual data scaling. We attribute this to the limited quality of synthetic data for pretraining Unlike standard translation tasks, the heterogeneity and long length of pretraining

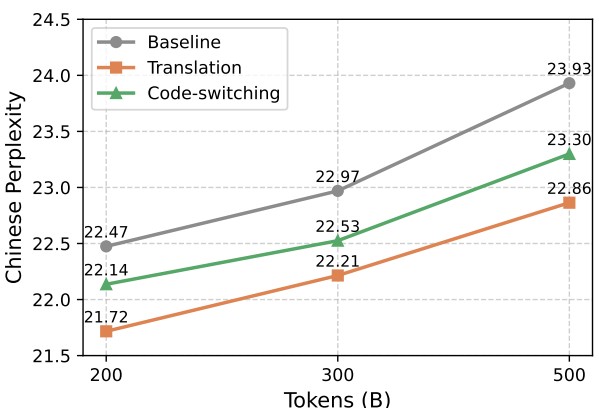

Figure 1: Comparison of translation and code-switching with baseline on Chinese PPL (↓). showing mitigation of degradation but fail to achieve cross-lingual data scaling.

text (up to 4,096 tokens in our setup) lead to translated outputs of considerably lower quality than native Chinese data. While employing larger LLMs could yield modest improvements in translation quality, the gains remain insufficient to close the gap with native data, and the sharply increased computational cost further limits the scalability of this approach. Moreover, since Chinese is relatively well resourced compared to many other languages, these limitations are expected to be even more severe for truly low-resource languages, where open-source LLMs typically produce translations of considerably lower quality. Additional experiments and analyses are provided in Appendix A.1.

**Code-Switching.** We next consider code-switching as an alternative approach to generate synthetic pretraining data for low-resource language. Following Wang et al. (2025b), we randomly select 20% of the English samples and replace 30% of their content with Chinese, either at the token or sentence level. In contrast to translation, code-switching modifies only partial tokens or sentences, thereby lowering the generation difficulty, but simultaneously constraining the maximum proportion of Chinese tokens present in the resulting pretraining data. As shown in Figure 1, code-switching attenuates the decline in Chinese performance but still fails to realize cross-lingual data scaling. Further experiments and analyses are provided in Appendix A.2.

Overall, the experiments demonstrate that existing data transformation techniques, including translation and code-switching, fail to produce synthetic data capable of sustaining cross-lingual data scaling. Moreover, generating such data incurs substantial computational overhead, posing a major obstacle to their use in large-scale pretraining. These limitations motivate us to explore knowledge transfer mechanisms as a more scalable alternative for cross-lingual data scaling.

## 3.4 KNOWLEDGE TRANSFER FOR CROSS-LINGUAL DATA SCALING

**Order Adjustment.** When data transformation fails to support cross-lingual data scaling, we turn to exploring how directly scaling high-resource language data can enhance LLM performance in low-resource languages. As shown in Figure 2a, directly scaling English data (Basline) not only fails to improve Chinese performance but actually leads to a consistent decline. We attribute this phenomenon to the decreasing proportion of Chinese data in the pretraining dataset as the amount of English data continues to increase. Inspired by previous work (Zheng et al., 2024; Bari et al.,

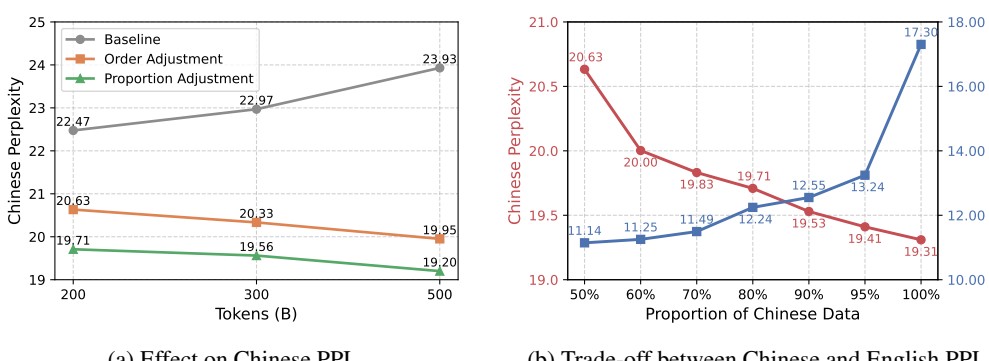

(a) Effect on Chinese PPL          (b) Trade-off between Chinese and English PPL

Figure 2: Effect of order and proportion adjustment on cross-lingual data scaling. **(a)** Chinese PPL (↓) as English data increases, showing that both adjustments enable cross-lingual data scaling. **(b)** Impact of varying the Chinese data proportion in the second stage, illustrating the trade-off between Chinese and English PPLs.

2025) that incorporates additional languages through continual pretraining, we investigate adjusting the language order during pretraining as a potential solution. Specifically, we adopt a two-stage pretraining paradigm, where the first stage consists solely of expanded English data, and the second stage involves a balanced mixture of English and Chinese data. Remarkably, as shown in Figure 2a, order adjustment enables Chinese performance to scale with the growth of English data. A key factor in this success is the appropriate learning rate schedule, which we analyze in more detail in Section 5.2.1.

**Proportion Adjustment.** Building on order adjustment, we further examine the effect of modifying the proportion of low-resource language data in the second stage. Since the total amount of low-resource language data is fixed, allocating a higher proportion necessarily shortens the duration of the second stage. As shown in Figure 2b, increasing the proportion of Chinese data substantially improves Chinese performance, but at the cost of a marked decline in English performance. From a practical perspective, we set the proportion of Chinese data to 80% in the second stage. In addition, Figure 2a shows that adjusting the proportion of low-resource data enables more effective cross-lingual data scaling.

Overall, the experiments demonstrate that adjusting the language order and proportion can efficiently achieve cross-lingual data scaling. These findings highlight the potential of cross-lingual knowledge transfer as a scalable solution to improve low-resource language performance, while simultaneously maintaining the sustained improvement of high-resource language performance. Based on proportion adjustment, we propose a new pretraining framework in the following section to further enhance cross-lingual data scaling.

## 4 SCALEX: SCALABLE CROSS-LINGUAL PRETRAINING FRAMEWORK

We aim to design a pretraining framework that enables cross-lingual data scaling. To this end, we propose **ScaleX**, which allows LLM performance in low-resource languages to scale with the dataset size of high-resource languages. Figure 3 provides an overview of ScaleX, depicting the data schedule and the corresponding learning rate schedule. The framework can be instantiated in two variants: a standard configuration without data repetition and an extended configuration that incorporates data repetition to further increase the exposure of low-resource language data.

ScaleX adopts a two-stage training paradigm, as illustrated in Figure 3a. In the first stage, the model is pretrained exclusively on expanding high-resource language data under a constant learning rate schedule. In the second stage, pretraining continues on a mixture of high- and low-resource languages under a cosine learning rate schedule, with the proportion of low-resource language data fixed at 80%. Because both the total amount of low-resource language data and the batch size are fixed, the number of training steps in the second stage is determined by the proportion of low-

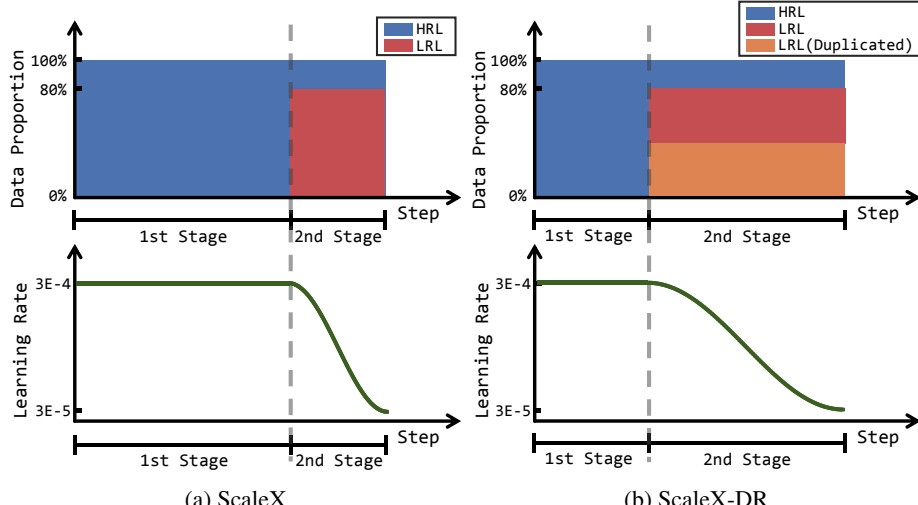

(a) ScaleX

(b) ScaleX-DR

Figure 3: Illustration of data and learning rate schedules in the two variants of ScaleX: **(a)** standard ScaleX; **(b)** ScaleX with data repetition (ScaleX-DR). "HRL" denotes high-resource languages and "LRL" denotes low-resource languages.

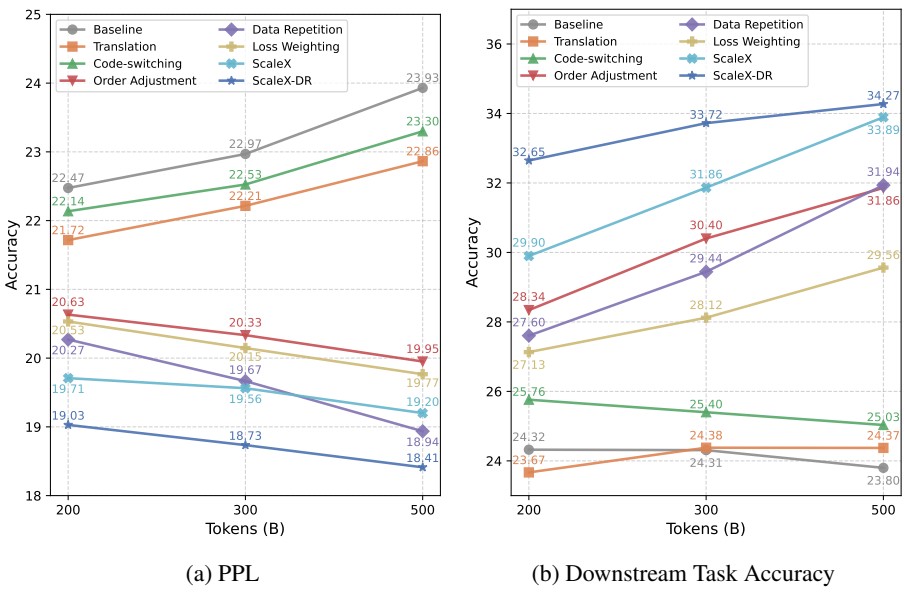

(a) PPL

(b) Downstream Task Accuracy

Figure 4: Comparison of ScaleX and ScaleX-DR with other approaches. **(a)** Chinese perplexity ($\downarrow$) as English data increases. **(b)** Average accuracy ($\uparrow$) over four Chinese downstream tasks as English data increases.

resource language data. Given this constraint, the number of steps in the first stage is naturally determined by the remaining high-resource language data.

Beyond the standard configuration, we further introduce a variant that incorporates data repetition, denoted as **ScaleX-DR**, as illustrated in Figure 3b. In this configuration, low-resource language data is duplicated, and an equal amount of high-resource language data is replaced to preserve dataset size. This adjustment shifts training steps from the first stage to the second stage, effectively increasing the number of updates on low-resource language data. Although the repetition count can be scaled to approximately 20 times (see Appendix A.3), in our experiments we adopt a single repetition setting, which is sufficient to demonstrate the complementary effect of data repetition within the ScaleX framework.

# 5 EXPERIMENTS

## 5.1 PERFORMANCE AND GENERALIZATION

**Main Results.** We compare ScaleX with other approaches in terms of PPL and downstream task accuracy. As shown in Figure 4, ScaleX and ScaleX-DR consistently achieve the best performance, with their advantage over other approaches widening as the amount of English data increases. Notably, both loss weighting and data repetition appear to exhibit scalability in improving low-resource language performance. However, their benefits are bounded: loss weighting yields largest gains with scaling factors of 10–20, while data repetition is most effective at 16–24 repetitions. Beyond these ranges, further scaling not only fails to improve performance but can even degrade it. Appendix A.3 provides detailed results on the scalability limits of these approaches

**Generalization in Multilingual Pretraining.** We evaluate the generalization of ScaleX in multilingual pretraining. Following prior work (Zhang et al., 2025; Wang et al., 2025b), we extend the bilingual setup by incorporating three additional low-resource languages: Turkish, Hungarian, and Bengali. In this setting, English serves as the high-resource language, while the other four are treated as low-resource languages. To simulate cross-lingual data scaling, we fix the dataset sizes of all low-resource languages and vary English data across three scales: 50B, 150B, and 250B tokens. The language statistics for these experiments are provided in Appendix A.4.

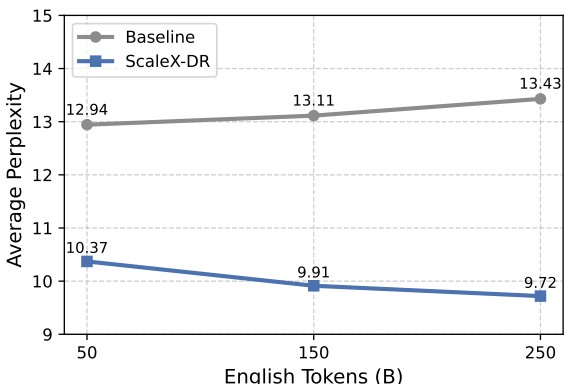

Figure 5: Comparison between ScaleX-DR and basline in multilingual pretraining.

As shown in Figure 5, ScaleX-DR consistently improves performance across all low-resource languages as the amount of English data increases, and its advantage over baseline becomes more pronounced at larger scales. These results demonstrate the generalization of ScaleX in multilingual pretraining. Moreover, the experiment shows that ScaleX continues to enable cross-lingual data scaling even under extreme imbalance, where the ratio of high- to low-resource data reaches 1:300–1:1500 and some low-resource datasets contain as few as 0.17B tokens.

**Generalization in Large-scale Pretraining.** We further evaluate the generalization of ScaleX in large-scale pretraining. Specifically, we train 2.5B LLMs on bilingual datasets of 500B and 1T tokens. The Chinese dataset is fixed at 200B tokens, constructed by duplicating 100B unique Chinese tokens once, while the English datasets contain 300B and 800B tokens, respectively. Owing to the prohibitive cost of large-scale pretraining, we do not provide comparisons with other approaches at this scale.

Table 1: Chinese performance of ScaleX-DR in large-scale pretraining.

| Language | Metric | Tokens | |
| --- | --- | --- | --- |
| | | 500B | 1T |
| Chinese | PPL | 15.45 | **15.20** |
| | ACC | 45.35 | **48.41** |
| English | PPL | 10.18 | **9.89** |
| | ACC | 37.47 | **42.39** |

As shown in Table 1, ScaleX delivers consistent improvements in both perplexity and downstream task performance across Chinese and English. These results highlight not only the scalability of ScaleX but also its applicability to practical large-scale pretraining.

## 5.2 ANALYSIS

### 5.2.1 EFFECT OF LEARNING RATE SCHEDULE

To examine the role of learning rate schedule in the scalability of ScaleX, we evaluate three configurations within the two-stage pretraining paradigm: (i) **Cos**, a conventional cosine learning rate

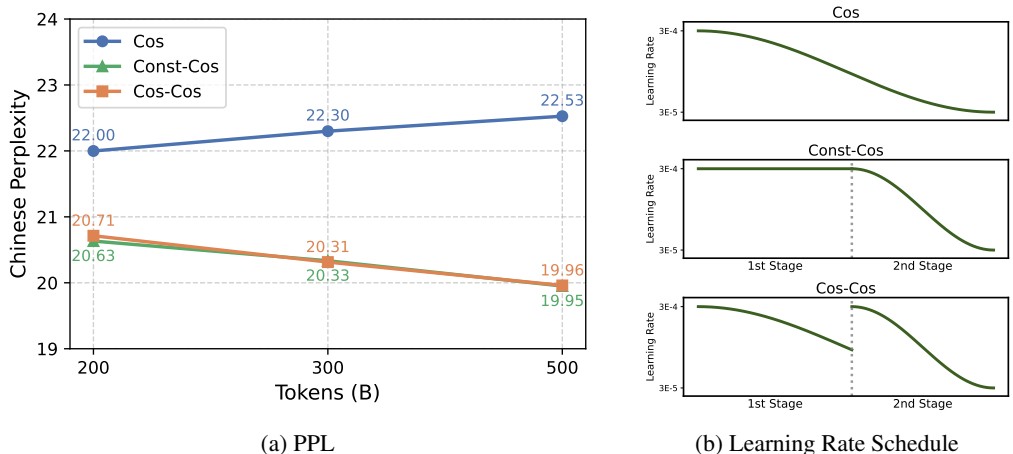

(a) PPL                    (b) Learning Rate Schedule

Figure 6: Effect of learning rate schedule on the scalability of ScaleX. **(a)** Chinese PPL as English data increases. **(b)** The corresponding learning rate schedules. Results highlight that maintaining a sufficiently large learning rate in the second stage is critical for preserving scalability of ScaleX.

schedule; (ii) **Const-Cos**, a constant learning rate schedule in the first stage, followed by a cosine learning rate schedule in the second stage; (iii) **Cos-Cos**, a reset cosine learning rate schedule, with the first stage following the initial part of the cosine learning rate schedule of (i) and the second stage adopting the cosine learning rate schedule from (ii).

As shown in Figure 6, ScaleX under the second and third learning rate configurations maintains cross-lingual scalability. We attribute this to the larger learning rate of the second stage, which is crucial for enhancing low-resource language performance. Our experiments indicate that the sustained improvements of ScaleX on low-resource languages are ultimately driven by the increasing amount of data in the first stage. However, before high-resource language data reaches a sufficiently large scale, the primary contributor to low-resource language performance remains the low-resource language data in the second stage. In this setting, if the learning rate in the second stage is too small, the contribution of low-resource language data diminishes significantly. When applying a conventional cosine learning rate schedule, the learning rate in the second stage decreases as the first-stage training lengthens, which in turn leads to a decline in low-resource language performance as high-resource language data increases. Therefore, maintaining a sufficiently large learning rate in the second stage is essential for preserving cross-lingual scalability of ScaleX.

### 5.2.2   EFFECT OF SHARED TOKENS

While the learning rate experiments clarify the complementary roles of the two pretraining stages, a key question remains: *why does scaling high-resource language data in the first stage consistently improve performance on low-resource languages?* Our statistical analysis shows that low-resource language samples, such as Chinese, contain a substantial number of English tokens. Beyond code-switching tokens, many samples share tokens across languages, including punctuation, numerals, and other symbols. This observation motivates the hypothesis that the performance gains of low-resource languages from the first stage stem from these shared tokens across languages.

To verify this hypothesis, we categorize the tokens in the bilingual dataset into three groups: Chinese, English, and others. Based on these categories, we construct datasets with controlled token types and design two-stage pretraining experiments, where the first stage uses only Chinese data and the second stage uses only English data. In this setup, Chinese is treated as the high-resource language and English as the low-resource language. This choice is motivated by the fact that fewer than 0.6% of Chinese samples are free of English tokens; if Chinese is designated as the low-resource language, it will be difficult to construct controlled experiments without compromising the quality of the low-resource language data.

We conduct four controlled experiments with different shared token configurations, as listed in Table 2. The results show that ScaleX exhibits scalability whenever at least one token category is

Table 2: Effect of different shared token settings on the scalability of ScaleX. In the Shared Token columns, "✓" and "✗" indicate the presence or absence of each token category across pretraining stages, while in the Scalable column they denote whether ScaleX exhibits scalability. English perplexities are reported for training data of 4B, 40B, and 80B tokens, highlighting the effect of scaling.

| # | Shared Token | | | Scalable | English Perplexity | | |
|---|---|---|---|---|---|---|---|
| | Chinese | English | Other | | 4B | 40B | 80B |
| 1 | ✗ | ✗ | ✗ | ✗ | **12.54** | 12.63 | 13.02 |
| 2 | ✗ | ✓ | ✗ | ✓ | 12.26 | 11.50 | **11.34** |
| 3 | ✗ | ✗ | ✓ | ✓ | 12.42 | 11.74 | **11.49** |
| 4 | ✗ | ✓ | ✓ | ✓ | 12.16 | 11.34 | **11.16** |

shared across stages. This finding suggests that shared tokens play a critical role in facilitating knowledge transfer between languages, thereby enabling cross-lingual scalability. Further analyses are provided in Appendix A.5.

## 6 CONCLUSION AND FUTURE WORK

To overcome the limitations of small and stagnant dataset sizes in low-resource languages, we introduce cross-lingual data scaling, which leverages high-resource language data to sustain performance gains for low-resource languages. Our experiments demonstrate that, while translation and code-switching are insufficient to achieve cross-lingual data scaling, adjusting language order and proportion can effectively realize it through knowledge transfer. Building on these findings, we propose ScaleX, a two-stage pretraining framework that enables LLM performance in low-resource languages to scale with the dataset size of high-resource languages. ScaleX consistently outperforms other approaches, with its advantage widening as high-resource language data increases, and further generalizes to both multilingual and large-scale bilingual pretraining. Finally, our analyses highlight learning rate schedule and shared tokens across languages are crucial for sustaining cross-lingual data scaling, offering guidance for future research on low-resource language modeling.

Future work can advance along two complementary directions. The first is to deepen the understanding of cross-lingual transfer mechanisms at a more fine-grained level, thereby establishing stronger theoretical foundations and guiding more principled pretraining designs. The second is to extend cross-lingual data scaling beyond text, for example to multimodal pretraining, thus broadening its applicability to more diverse real-world scenarios.

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

## A APPENDIX

### A.1 IMPLEMENTATION OF TRANSLATION

To balance translation difficulty and generation cost, we design a segmentation-based translation pipeline, which splits long texts into shorter segments to reduce translation complexity and cost. The

pipeline consists of three steps: (i) Segmentation. Training samples are segmented into sentences using BlingFire[1]. (ii) Translation. Segmented sentences are translated using Qwen2.5-Instruct series models, executed with the vLLM framework (Kwon et al., 2023). (iii) Merging. Translated sentences are concatenated in their original order to form synthetic low-resource language documents used for pretraining.

On the other hand, an appropriate trade-off between translation quality and generation cost requires selecting a suitably sized open-source LLM for translation. We evaluate synthetic data quality using PPL and measure cost by the GPU hours required to generate 10B tokens. As shown in Table 3, larger Qwen models yield higher-quality translations but also incur substantially higher costs. When the model size exceeds 7B parameters, the improvements in data quality are marginal, whereas the computational overhead grows substantially. Therefore, we adopt Qwen2.5-7B-Instruct as our translation model.

Table 3: Data quality and computational cost of different Qwen2.5-Instruct models. Data quality is measured by PPL, and cost is measured by GPU hours required to generate 10B tokens.

| Model | Perplexity | GPU Hours |
|---|---|---|
| Qwen2.5-1.5B | 34.45 | 810 |
| Qwen2.5-3B | 35.76 | 979 |
| Qwen2.5-7B | 30.46 | 1,696 |
| Qwen2.5-14B | 28.31 | 2,640 |
| Qwen2.5-32B | 28.47 | 6,636 |
| Native Chinese | 11.58 | - |

It is worth noting that even with Qwen-32B, the quality of synthetic Chinese data generated via translation remains substantially below that of native Chinese. Given that Chinese is relatively well resourced compared to many other languages, this gap is likely to be even larger for truly low-resource languages, where open-source LLMs tend to produce translations of lower quality.

## A.2 Implementation of Code-switching

In our bilingual experiments, we adopt SynCS (Wang et al., 2025b) to introduce Chinese content into English samples. Specifically, we randomly select 20% of the English samples and replace 30% of their content with Chinese, either at the token or sentence level. The 20% sampling ratio follows the original SynCS setup, while the 30% replacement ratio is empirically selected as the most effective configuration. Additionally, the code-switched data are generated using a fine-tuned Qwen2.5-3B-Instruct model trained on the Chinese–English code-switching dataset released with SynCS.

To study the effect of replacement ratios on code-switching, we train 1.3B LLMs on 200B tokens, varying the ratio across 10%, 30%, 50%, 70%, and 90%. As shown in Table 4, simply increasing the proportion of Chinese tokens does not lead to consistent improvements in Chinese performance, suggesting that there is no straightforward benefit from further intensifying the code-switching ratio. Moreover, compared to generating pretraining data for low-resource languages via translation, code-switching entails similarly high generation cost and adds the extra burden of LLM fine-tuning.

Table 4: LLM performance across different code-switching replacement ratios.

| Ratio | Perplexity | Accuracy |
|---|---|---|
| 10% | 22.3 | 24.7 |
| **30%** | 22.1 | **25.8** |
| 50% | **22.0** | 24.4 |
| 70% | 22.1 | 23.7 |
| 90% | 21.9 | 25.2 |

## A.3 Scalability of Data Repetition and Loss Weighting

To further examine the scalability of loss weighting, we conduct supplementary experiments with 1.3B LLMs trained on 1T tokens. For loss weighting, as shown in Figure 7a, we observe that a model trained on 500B tokens with a $10\times$ weight outperforms a model trained on 1T tokens with a $20\times$ weight. This suggests the presence of a threshold in the $10$–$20\times$ range, beyond which increasing the loss weight no longer yields benefits and can even degrade performance.

On the other hand, even at the 1T-token scale, the LLMs trained with data repetition show no clear performance plateau in Chinese. Due to the prohibitive cost of training at the trillion-token scale, we instead examine the upper bound of data repetition at a smaller scale. Here, the Chinese dataset

---

[1]https://github.com/Microsoft/BlingFire

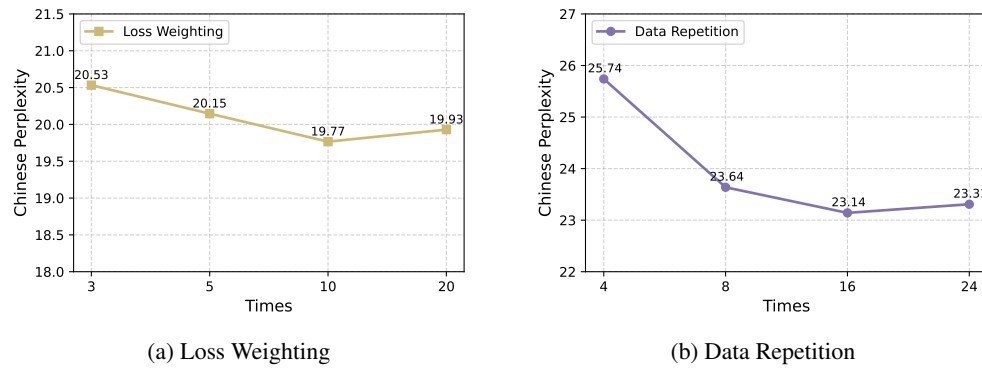

(a) Loss Weighting  (b) Data Repetition

Figure 7: Scalability of loss weighting and data repetition.

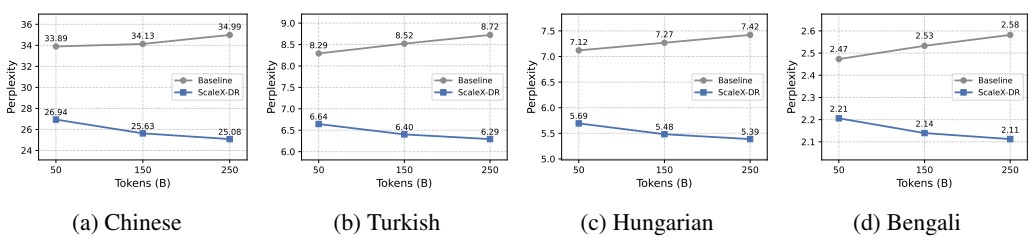

(a) Chinese  (b) Turkish  (c) Hungarian  (d) Bengali

Figure 8: The detailed performance of different low-resource languages in Figure 5.

is derived from 5B unique tokens and expanded through repetition, while the English dataset is enlarged with new data to keep the two languages balanced. As shown in Figure 7b, repeating the limited Chinese data $24\times$ yields lower performance than that of $16\times$, suggesting that the benefits peak between $16\times$ and $24\times$. These findings indicate that, similar to loss weighting, data repetition enhances performance only up to a certain point, after which additional repetitions result in degradation.

## A.4 IMPLEMENTATION OF MULTILINGUAL PRETRAINING

Table 5: Language statistics in the multilingual pretraining setup. The column "LRL" specifies whether the corresponding language is considered as low-resource language.

| Language | Language Family | LRL | Tokens (B) | Proportion (%) | |
|---|---|---|---|---|---|
| | | | | 50B | 250B |
| English | Indo-European | ✗ | 50–250 | 89.23 | 97.64 |
| Chinese | Sino-Tibetan | ✓ | 3.99 | 7.12 | 1.56 |
| Turkish | Turkic | ✓ | 1.12 | 2.00 | 0.43 |
| Hungarian | Uralic | ✓ | 0.76 | 1.35 | 0.30 |
| Bengali | Indo-European | ✓ | 0.17 | 0.30 | 0.07 |

Table 5 reports the language statistics for the multilingual pretraining experiments. As English data increases, the proportions of other languages scale accordingly, with the 50B English configuration following the distribution in the CulturaX dataset (Nguyen et al., 2024). On the other hand, Figure 8 presents the performance of individual low-resource languages in Figure 5.

## A.5 IMPLEMENTATION OF SHARE TOKEN

To examine the effect of shared tokens across languages on ScaleX, we first categorize tokens into three groups. Using this categorization, we construct bilingual datasets with different token compo-

Table 6: Proportion of bilingual samples containing different types of token.

| Zh | En | Ot | Proportion | |
|---|---|---|---|---|
| | | | Chinese | English |
| ✓ | ✓ | ✓ | 98.20% | 2.80% |
| ✓ | ✗ | ✓ | 1.80% | 0.00% |
| ✗ | ✓ | ✓ | 0.00% | 97.20% |

Table 7: Effect of different shared token settings on the scalability of ScaleX. "✓" and "✗" respectively indicate whether a token category is shared across stages or whether ScaleX is scalable.

| # | Shared Token | | | Scalable | English Perplexity | | |
|---|---|---|---|---|---|---|---|
| | Zh | En | Ot | | 4B | 40B | 80B |
| 1 | ✗ | ✗ | ✗ | ✗ | 12.54 | 12.63 | 13.02 |
| 2 | ✓ | ✗ | ✗ | ✗ | 12.59 | 12.55 | 13.01 |
| 3 | ✗ | ✓ | ✗ | ✓ | 12.26 | 11.50 | 11.34 |
| 4 | ✗ | ✗ | ✓ | ✓ | 12.42 | 11.74 | 11.49 |
| 5 | ✗ | ✓ | ✓ | ✓ | 12.16 | 11.34 | 11.16 |
| 6 | ✓ | ✓ | ✓ | ✓ | 12.18 | 11.36 | 11.17 |

sitions, which serve as the basis for evaluating the impact of shared tokens. Specifically, the tokens are categorized into three groups as follows:

- **Chinese tokens (Zh)**: consisting of characters in the CJK Unified Ideographs block, primarily in the Unicode range U+4E00–U+9FFF.

- **English tokens (En)**: consisting solely of the Latin alphabet in Unicode ranges U+0041–U+005A and U+0061–U+007A.

- **Other tokens (Ot)**: tokens excluding English and Chinese tokens, such as punctuation, digits, special symbols, and other symbols.

Based on these definitions, we further classify bilingual samples according to the token types they contain. As shown in Table 6, about 98.20% of Chinese samples contain all three token types, whereas only 1.8% contain Chinese and other tokens without English. This imbalance poses a practical issue: to obtain sufficient samples containing only Chinese and other tokens (but no English), one needs to strip English tokens from the majority of Chinese samples that contain all three token types. Such token removal perturbs the Chinese data, thereby introducing preprocessing-induced confounders and undermining the validity of any conclusions about the model's performance on Chinese. In contrast, applying analogous token removal to the high-resource language in the first pretraining stage has a much smaller effect, since evaluation focuses on the low-resource language. Therefore, without loss of generality, we designate Chinese as the high-resource language and English as the low-resource language in this experiment. Accordingly, the LLM is pretrained exclusively on Chinese data in the first stage, followed by English data in the second stage.

To further investigate the role of shared tokens, we design six controlled experiments by varying which token categories (Zh, En, Ot) are shared across the two pretraining stages. The detailed experimental settings are summarized in Table 7. We observe that ScaleX remains scalable when the shared tokens correspond to those required by the low-resource language in the second stage (e.g., English or Other tokens). In contrast, scalability does not hold when the shared tokens belong to categories unused by the low-resource language in the second stage (e.g., Chinese tokens). Moreover, the performance gains in #5 and #6 are substantially larger than in #3 and #4, suggesting that a greater degree of token sharing tends to yield stronger improvements in low-resource language performance.

### A.6 Use of LLMs

We make limited use of LLMs only as an auxiliary tool for proofreading and minor language polishing, including grammar checking, wording refinements, and terminology consistency. LLMs do not contribute to the conceptualization, experimental design, implementation, or analysis of this research, and all scientific content and contributions are the work of the authors.

