# OpenReview forum: "Cross-Lingual Data Scaling for Large Language Models"
_ICLR.cc/2026/Conference — Submitted to ICLR 2026_

### Official Review · Reviewer_77Sf · 2025-10-27

**Soundness:** 2
**Presentation:** 3
**Contribution:** 2
**Rating:** 4
**Confidence:** 4

**Summary:**

The authors analyze different approaches to scale low-resource language
performance by scaling the dataset size of high-resource languages: (i)
data transformation (creating low-resource synthetic data from the
high-resource language data), and (ii) knowledge transfer. Motivated by
the finding that knowledge transfer outperforms data transformation
techniques, the authors propose ScaleX, a pretraining framework where
models are first trained only on the high-resource language, and are
then trained on a mixture of high and low-resource language data. ScaleX
outperforms baseline approaches that rely on synthetic data.

**Strengths:**

- The authors find that knowledge transfer techniques can leverage
high-resource language data to improve low-resource performance. Their
approach outperforms methods that rely on synthetic data generation.
- The preliminary studies that motivate the design of ScaleX are
interesting and well-structured.

**Weaknesses:**

- The authors choose Chinese, a class-5 language, to treat as a
"low-resource" language across the majority of experiments. While I
understand that higher availability of data facilitates data sampling
and the design of the experimental setup, I believe the experiments in
sections 3.3 and 3.4 would benefit from being repeated with truly
low-resource languages. Specifically, the authors state that
"translation alleviates performance degradation compared to baseline,
but Chinese performance still declines as English data increases". Would
this also be true for low-resource languages, across language families,
and across translation models? To make the case for the usage of this
framework on truly low-resource language performance improvement, I
would suggest deepening these analyses.

- The authors evaluate ScaleX on Turkish, Hungarian, and Bengali. A
closer look at the appendix shows that perplexity decreases very
slightly across the size of English data, notably on Bengali. Are these
results statistically significant?

**Questions:**

- The sections on the effects on learning rate schedule and shared
tokens are extremely interesting. Are the results of these experiments
similar across other language pairs?
- Typo: "Basline" (line 213)

---

> ### Author Response · Authors · 2025-12-03
>
> **Q1**. Why is Chinese treated as a “low-resource” language?
>
> Our choice of Chinese as the target “low-resource” language is driven primarily by data quality and controllability.
> We use processed, high-quality LLM pretraining data, whose cleanliness and consistency are not achievable with typical open-source corpora. High-quality data better reflects real pretraining settings, reduces noise-induced confounding effects, and leads to faster and more stable training convergence.
> In addition, our approach is language-agnostic, which is also verified empirically through our multilingual experiments.
> Finally, several recent works have adopted the same experimental setup by treating Chinese as a low-resource language in LLM pretraining [1–2].
>
> [1] Breaking Language Barriers: Cross-Lingual Continual Pre-Training at Scale, EMNLP 2024
>
> [2] Investigating and Scaling up Code-Switching for Multilingual Language Model Pre-Training, ACL Findings 2025
>
>
> **Q2**. Would the conclusions generalize to other low-resource languages, across language families or MT models?
>
> Yes. Given the data-quality considerations and the high cost of large-scale pretraining, we conduct our main analysis using Chinese data and then apply the findings to other low-resource languages.
> In multilingual pretraining experiments, we intentionally select languages from diverse families, including Chinese (Sino-Tibetan), Turkish (Turkic), Hungarian (Uralic), and Bengali (Indo-European).
> Since our method is driven primarily by optimization rather than language-specific techniques, and this is further supported by the multilingual pretraining results reported in our paper, we believe that ScaleX has strong generalization across languages.
>
> In practice, machine translation is not an effective solution for cross-lingual data scaling.
> When shifting from Chinese to truly low-resource languages, translation quality deteriorates substantially, leading to synthetic corpora that are far inferior to real training data.
> By contrast, as shown in [3], when the repetition factor is small (e.g., <10×), repeated data retain utility comparable to real data.
> If machine translation is expected to match the effectiveness of data repetition and provide pretraining signals comparable to real data, then translation quality for low-resource languages would need to reach very high levels.
>
> [3] Scaling Data-Constrained Language Models, NeurIPS 2023
>
>
> **Q3**. Perplexity decreases only slightly on Bengali. Are the gains statistically significant?
>
> Yes. Perplexity is defined as an exponential function of cross-entropy, improvements become progressively less visible on the PPL scale as cross-entropy decreases, even when the underlying loss reduction is meaningful.
>
> | **Language**  | **PPL (50B)** | **PPL (250B)** |Δ |
> | ------------- | --------------| ---------------| ------|
> | **Chinese**   | 26.94         | 25.08          | –1.86 / 6.90% |
> | **Turkish**   | 6.64          | 6.29           | –0.35 / 5.27% |
> | **Hungarian** | 5.69          | 5.39           | –0.30 / 5.27% |
> | **Bengali**   | 2.21          | 2.11           | –0.10 / 4.52% |
>
> As shown in the table above, when PPL is high, reductions in PPL appear more pronounced.
> Furthermore, the percentage decline of PPLs in various languages is around 5%.
>
>
> **Q4**. Do learning-rate scheduling and shared-token effects generalize to other language pairs?
>
> Yes. Both phenomena arise from model optimization dynamics and vocabulary interactions, rather than properties of any particular language.
> Our multilingual experiments already demonstrate that the behavior persists across different language families. This further supports that our proposed methods and analyses are language-independent and apply broadly to multilingual pretraining setups.

---

### Official Review · Reviewer_d6jm · 2025-10-31

**Soundness:** 2
**Presentation:** 3
**Contribution:** 2
**Rating:** 2
**Confidence:** 5

**Summary:**

The paper proposes to train models in a targeted low resource language with the help of data from an auxiliary language. The experiments consider (i) transformations of the auxiliary data (translation, code mixing), (ii) varying the mixing proportion of aux/target language and (iii) pretraining on auxiliary data followed by joint training. The paper concludes that (iii) is the best strategy with experiments over Chinese (low resource) + English (auxiliary).

**Strengths:**

The paper reads well, it defines the problem clearly and proposes a sensible list of methods to benefit from auxiliary language data.

**Weaknesses:**

1. The paper does not compare with strategies established in recent papers (even when these papers are cited, e.g. for the first two).


He et al., 2024 (Findings of ACL) — Scaling Laws for Multilingual Language Models: derives scaling relations that include language-family sampling ratios; directly relevant to “how much” auxiliary language to mix. arXiv:2410.12883
-> why not determine the mixing weights with scaling laws as in this paper?


Li et al., 2024 (NAACL) — Upsample or Upweight? Balanced Training on Heavily Imbalanced Data Mixtures: analyzes temperature sampling vs. scalarization and proposes “Cooldown,” useful for tuning bilingual/multilingual mixtures. arXiv:2410.04579
-> why not try the mixing schedule from this paper?


Seto et al., 2024 (Findings of ACL) — Training Bilingual LMs with Data Constraints in the Targeted Language. Summarizes auxiliary-language vs. translate-then-train strategies and practical upsampling. arXiv:2411.12986
-> why not tune a mixing weight when introducing translation data?




2. The paper targets Chinese end-task performance but intermediate results are established with Chinese perplexity. It remains to be shown that a degradation in perplexity always translates into a degradation in end-task performance. It would be more sensible to split the tasks into validation and test tasks in order to drive data decisions based on end-task validation performance. E.g. I would not be surprised if machine translation improves end-task performance but degrades perplexity in line with [Seto et al., 2024]. If one is interested only in perplexity, Figure 2b. suggests training on 100% Chinese data, yet you pick 80% in practice.


3. The paper defines a search space that is only partially searched. The paper considers that (i) the auxiliary data could be introduced with 3 types of transformation (none, code switching, translation), (ii) the auxiliary data mixing proportion is important, (iii) the auxiliary data can be introduced before or after pretraining on auxiliary data. Yet the conclusions of the different parts are not connected together. It seems that Figure 1 suggests that translating the data is better than using the English data as-is, yet the next experiments (mixing, continued pretraining) do not use translation data. Similarly the mixing experiments do suggest to use only Chinese data in the second phase (Figure 2b) yet the next experiments consider different proportions.


4. The paper considers a single language pair, and only two model scales (1B, 2.5B). The conclusions of the paper cannot be applied to other language pairs and different model sizes. It would be helpful to consider more language and more (smaller) model scales to draw conclusions that can be more generic in terms of language proximity, translation performance and model sizes. Similarly only two sizes of Chinese set are considered (50B and 100B tokens) are considered, what happens if less tokens are available in the targeted language? How does the performance scale wrt to the number of target tokens? How to select the fraction of English data in the second phase when the amount of Chinese data changes?

**Questions:**

1. Why did you pick 80% as the mixing weight for the second phase?
2. For the learning rate schedule, does the training schedule include a warmup phase? Is the warmup applied again at the beginning of the second phase? Is cos-cos stopping at 1/2 max lr for the first phase? Why?
3. Why not use translated data (En->Zh) for phase 1 and phase 2?
4. Why not include a baseline with 100% Chinese data with varying sizes? This would help evaluate the value of adding auxiliary (En) data (possibly translated) as opposed to genuine Chinese data.
5. What is the purpose of the English end-tasks? They are only used in Table 1 to say that better results are achieved when more English data is available. I would remove them.
6. Do 7a and 7b (impact of loss weighting and repetition) behave differently for end-task performance?

---

> ### Author Response · Authors · 2025-12-03
>
> **Q1**. The paper does not compare with strategies established in recent three papers (even when these papers are cited, e.g. for the first two).
>
> 1.The mixing ratios in “Scaling Laws for Multilingual LMs” optimize average multilingual performance, whereas our goal is to improve specific low-resource languages in a two-phase setting. Because the scaling-law assumptions do not match our problem, its ratios are not applicable here.
>
> 2.The method in Li et al. (2024) is designed for balancing multilingual joint training and improving average performance across languages, whereas our setting fixes the low-resource data and studies cross-lingual scaling in a two-phase framework. Since their assumptions and objectives differ fundamentally from ours, their mixing schedule is not applicable to our problem.
>
> 3.Seto et al. (2024, Findings of ACL) focuses primarily on improving the use of translated data. As discussed in our response to Q7, translation can alleviate performance degradation but cannot produce performance improvements. Enhancing translation quality or applying improved translation strategies would only reduce the degree of degradation and would not affect our core conclusion.
>
> We emphasize that the central question of our paper is whether high-resource language data can improve low-resource language performance, not how large the improvement is or how much the degradation can be mitigated.
> Most techniques mentioned in the reviews, including various mixing strategies or their variants, primarily affect the magnitude of gains or losses rather than the fundamental property of whether cross-lingual data scaling is achievable.
> In other words, exploring these additional, computation-heavy pretraining variants would not change the core conclusion of the paper.
>
>
> **Q2**. The paper targets Chinese end-task performance but intermediate results are established with Chinese perplexity. It remains to be shown that a degradation in perplexity always translates into a degradation in end-task performance. If one is interested only in perplexity, Figure 2b. suggests training on 100% Chinese data, yet you pick 80% in practice.
>
> Based on our extensive pre-training experience, perplexity and downstream task performance exhibit highly consistent trends in the vast majority of cases.
> One notable exception is when the model size is very small (e.g., <1B parameters). In such settings, the model’s downstream accuracy is often too low to be meaningful, and fluctuations may appear random even when PPL improves. This is a well-known limitation of small models rather than a contradiction between PPL and downstream behavior.
>
> Regarding the choice of 80% Chinese data in the second phase:
> the decision was driven by practical deployment considerations, where retaining a certain level of English capability was desirable. As the reviewer correctly notes, using 100% Chinese data would further strengthen the improvement for Chinese. In the revised version of the paper, we will update the experiment to report results with 100% Chinese data; this change does not alter any conclusions and only enhances the reported trends.
>
>
> **Q3**. The paper defines a search space that is only partially searched.
>
> In the early stage of our study, we extensively explored translation, as well as various combinations of other techniques. However, our experiments consistently showed that data repetition substantially outperforms both translation and code-switching, and therefore we did not continue to build subsequent experiments around these two transformations.
>
> Our goal is to understand how high-resource language data can be leveraged to improve low-resource language performance, not merely mitigate degradation. As shown in the paper, although translation and code-switching can reduce the rate of performance drop, they fail to produce a positive upward trend. Importantly, this limitation persists even when combining these techniques with other strategies.
>
> Whenever modifying data is required, repetition is a more reliable and cost-effective option than translation or code-switching. Repetition is the only method among the three that consistently enables performance improvements within a controlled range of repetition factors.
>
> Given the substantial pre-training cost of large language models, we focused our subsequent analysis on repetition-based approaches rather than exploring additional variants or combinations of translation and code-switching, which our preliminary results already indicated would be suboptimal.

---

> ### Author Response · Authors · 2025-12-03
>
> **Q4**. The paper considers a single language pair, and only two model scales (1B, 2.5B). The conclusions of the paper cannot be applied to other language pairs and different model sizes. It would be helpful to consider more language and more (smaller) model scales to draw conclusions that can be more generic in terms of language proximity, translation performance and model sizes.
>
> Based on our experience, the results obtained on 1B and 2.5B LLMs are consistent, suggesting that the method is stable across this sub-3B scale range. This provides practical evidence for the method’s applicability in commonly used mid-size LLMs.
> When LLMs are smaller than 1B, it is common to observe decreasing PPL accompanied by highly unstable downstream-task performance. Such fluctuations introduce confounding factors that make it difficult to draw reliable conclusions.
>
> In Section 5.1 (“Generalization in Multilingual Pretraining”), we conduct experiments on English, Chinese, Turkish, Hungarian, and Bengali, where the latter four serve as low-resource languages from different language families. Their respective data scales are 3.99B, 1.12B, 0.76B, and 0.17B tokens, and each experiment includes 50B, 150B, and 250B English tokens.
> The results are consistent with our bilingual findings, demonstrating that ScaleX generalizes robustly across languages, language families, and data scales.
>
>
> **Q5** .What happens if less tokens are available in the targeted language, How does the performance scale wrt to the number of target tokens? How to select the fraction of English data in the second phase when the amount of Chinese data changes?
>
> Our multilingual pretraining experiments in Section 5.1 (“Generalization in Multilingual Pretraining”) already provide partial answers to this question. The four low-resource languages included in the study span different data scales, and the results show that our conclusions hold across these varying resource levels.
> Our primary focus is the effect of increasing high-resource data on the performance of a low-resource language. If one simply increases the amount of low-resource data, then its performance naturally improves.
>
> As described in Section 4, the proportion used in Phase 2 is determined under a fixed data-mixing ratio. Since the size of the low-resource corpus is predetermined, fixing the ratio automatically determines the amount of high-resource data in Phase 2, and therefore the total number of training steps.
>
>
> **Q6**. For the learning rate schedule, does the training schedule include a warmup phase? Is the warmup applied again at the beginning of the second phase? Is cos-cos stopping at 1/2 max lr for the first phase? Why?
>
> All training runs in both Stage 1 and Stage 2 are conducted without a warm-up phase.
> Based on our empirical observations, removing warm-up leads to slightly better performance under our setting; therefore, we consistently disable warm-up across all experiments.
> For Stage 2, the model parameters have already been sufficiently optimized during Stage 1, so an additional warm-up phase is unnecessary. This observation is also consistent with prior findings [1].
>
> In Figure 6, the learning rate for the cos–cos schedule does not decay to its minimum value. This design choice ensures that its Phase-1 learning rate curve is identical to that of the cos (first setting in Figure 6), allowing a controlled comparison. By holding Phase-1 constant, we isolate the effect of Phase-2 learning rate design and show that it is crucial for achieving cross-lingual data scaling.
> This is mentioned in section 5.2.1 of the paper.
>
> [1] A Learning Rate Path Switching Training Paradigm for Version Updates of Large Language Models. EMNLP 2024.

---

> ### Author Response · Authors · 2025-12-03
>
> **Q7**. Why not use translated data (En->Zh) for phase 1 and phase 2? Why not include a baseline with 100% Chinese data with varying sizes, this would help evaluate the value of adding auxiliary (En) data (possibly translated) as opposed to genuine Chinese data.
>
> In fact, we have tried adding translation data to either Phase 1 or Phase 2, but the results were far inferior to those of our proposed ScaleX, consistent with our earlier finding that machine-translated data performs much worse than data repetition.
>
> In practice, machine translation is not an effective solution for cross-lingual data scaling.
> When shifting from Chinese to truly low-resource languages, translation quality deteriorates substantially, leading to synthetic corpora that are far inferior to real training data.
> By contrast, as shown in [2], when the repetition factor is small (e.g., <10×), repeated data retain utility comparable to real data.
> If machine translation is expected to match the effectiveness of data repetition and provide pretraining signals comparable to real data, then translation quality for low-resource languages would need to reach very high levels.
>
> 100% Chinese data of varying sizes does indeed help to observe the gap between the current translation method and the upper limit. However, this is not the focus of our research, so no experiments were conducted.
>
> [2] Scaling Data-Constrained Language Models, NeurIPS 2023
>
>
> **Q8**. What is the purpose of the English end-tasks? They are only used in Table 1 to say that better results are achieved when more English data is available. I would remove them.
>
> The English downstream tasks are included to assess the model’s English performance and to address a common concern: that PPL improvements may not translate into downstream gains (as raised by Reviewer 8jCx).
> To provide a more complete view, we will add additional English downstream results in the appendix of the revised manuscript.
>
>
> **Q9**. Do 7a and 7b (impact of loss weighting and repetition) behave differently for end-task performance?
>
> Similar to the PPL results, both methods exhibit a rise-then-drop pattern on downstream tasks, although the turning points differ between the two approaches.
> The important conclusion is that neither method can stably achieve cross-lingual data scaling.

---

### Official Review · Reviewer_JH1T · 2025-11-03

**Soundness:** 3
**Presentation:** 2
**Contribution:** 3
**Rating:** 6
**Confidence:** 3

**Summary:**

This paper investigates the impact of cross-lingual data scaling on capability transfer in LLMs. By comparing data conversion and knowledge transfer paradigms, the authors find that knowledge transfer enables low-resource languages to benefit from the scaling of high-resource language data. Building on this insight, they propose ScaleX, a two-stage pretraining framework for efficient cross-lingual data scaling, and demonstrate its effectiveness through extensive experiments.

**Strengths:**

1. This paper provides a clear and insightful analysis of cross-lingual data scaling mechanisms, offering direct guidance for optimizing LLM training strategies.

2. The experimental design is systematic, covering multiple languages across different resource levels with thorough validation.

3. The methodology is clearly described, and the overall structure of the paper is logical and well-organized.

**Weaknesses:**

1. Using Chinese as an example of a “ow-resource language is questionable, as its data scale far exceeds that of genuinely low-resource languages. As shown in Table 5, Chinese data (3.99B tokens) is much larger than Bengali (0.17B tokens).

2. The translation experiments only evaluate open-source single-step models (the Qwen2.5-Instruct series, as noted in Appendix A.1), without comparison to stronger closed-source models such as GPT-4o or Claude.

3. The code-switching experiments employ fixed mixing ratios (20% sample-level, 30% token-level replacement). Appendix A.2 tests only static ratios from 10%–90%, without exploring dynamically adjusted schedules.

4. The choice of 80% low-resource ratio in the second stage lacks empirical or theoretical justification.

5. Although ScaleX avoids high-cost translation operations, the two-stage training procedure itself may introduce additional scheduling complexity. A comparison of training time and resource consumption would strengthen the practical analysis.

6. Details regarding ScaleX-DR repetitions are missing. For example, the paper mentions a “single repetition setting” but does not specify repetition curves, and the claim that “benefits peak between 16x and 24x” lacks an explanation for the subsequent performance drop.

**Questions:**

Refer  Weaknesses

---

> ### Author Response · Authors · 2025-12-03
>
> **Q1**. Using Chinese as an example of a "low-resource language" is questionable.
>
> Our choice of Chinese as the target “low-resource” language is driven primarily by data quality and controllability.
> We use processed, high-quality LLM pretraining data, whose cleanliness and consistency are not achievable with typical open-source corpora. High-quality data better reflects real pretraining settings, reduces noise-induced confounding effects, and leads to faster and more stable training convergence.
> In addition, our approach is language-agnostic, which is also verified empirically through our multilingual experiments.
> Finally, several recent works have adopted the same experimental setup by treating Chinese as a low-resource language in LLM pretraining [1–2].
>
> [1] Breaking Language Barriers: Cross-Lingual Continual Pre-Training at Scale, EMNLP 2024
>
> [2] Investigating and Scaling up Code-Switching for Multilingual Language Model Pre-Training, ACL Findings 2025
>
>
> **Q2**. The translation experiments only evaluate open-source single-step models (the Qwen2.5-Instruct series, as noted in Appendix A.1), without comparison to stronger closed-source models such as GPT-4o or Claude.
>
> For LLM pretraining at trillion-token scale, using models such as GPT-4o or Claude to generate synthetic data is not feasible, as the associated cost would be prohibitively high.
> In practice, machine translation is not an effective solution for cross-lingual data scaling.
> When shifting from Chinese to truly low-resource languages, translation quality deteriorates substantially, leading to synthetic corpora that are far inferior to real training data.
> By contrast, as shown in [3], when the repetition factor is small (e.g., <10×), repeated data retain utility comparable to real data.
> If machine translation is expected to match the effectiveness of data repetition and provide pretraining signals comparable to real data, then translation quality for low-resource languages would need to reach very high levels.
>
> [3] Scaling Data-Constrained Language Models, NeurIPS 2023
>
>
> **Q3**. The code-switching experiments employ fixed mixing ratios (20% sample-level, 30% token-level replacement). Appendix A.2 tests only static ratios from 10%–90%, without exploring dynamically adjusted schedules.
>
> Similar to the Q2, code-switching is only one potentially feasible direction, and our experiments already show that its effectiveness is lower than that of machine translation.
> Given the high cost of large-scale pretraining, we did not further explore how to optimize this method for cross-lingual data scaling.
> We believe that small performance improvements would not change the fundamental nature of the method and would not suddenly enable it to achieve cross-lingual data scaling.
> Therefore, such variations would not affect the conclusions of the paper.
>
>
> **Q4**. The choice of 80% low-resource ratio in the second stage lacks empirical or theoretical justification.
>
> The choice of using an 80% low-resource ratio in the second stage was motivated by practical deployment considerations, where preventing excessive degradation of high-resource language performance is often important.
>
> This is a good suggestion. In the revised version of the paper, we will update the experiment to use a 100% low-resource ratio in the second stage. Since the paper focuses primarily on low-resource language performance, this modification does not affect our conclusions and would in fact further improve the low-resource results.
>
>
> **Q5**. Although ScaleX avoids high-cost translation operations, the two-stage training procedure itself may introduce additional scheduling complexity. A comparison of training time and resource consumption would strengthen the practical analysis.
>
> One major advantage of our method is that it does not introduce any additional pretraining cost.
> The only extra effort lies in data preparation, such as reordering the pretraining corpus or duplicating the low-resource language data.
> However, the cost of these preprocessing steps is negligible compared to the computational cost of large-scale LLM pretraining.

---

> ### Author Response · Authors · 2025-12-03
>
> **Q6**. Details regarding ScaleX-DR repetitions are missing. For example, the paper mentions a “single repetition setting” but does not specify repetition curves, and the claim that “benefits peak between 16x and 24x” lacks an explanation for the subsequent performance drop.
>
> As shown in Figure 3, we repeat the low-resource data only once.
> The performance of our ScaleX-DR can be further improved when the number of repetitions is increased.
> The statement that “benefits peak between 16× and 24×” refers to the baseline data repetition method, which exhibits diminishing or even negative returns when the repetition factor becomes excessively large. This observation is used to illustrate that relying solely on data repetition is not a viable solution, further highlighting the advantage of our proposed ScaleX framework.
> We will make this detail more explicit in the revised version of the paper.

---

### Official Review · Reviewer_8jCx · 2025-11-10

**Soundness:** 2
**Presentation:** 2
**Contribution:** 2
**Rating:** 4
**Confidence:** 4

**Summary:**

This paper studies cross-language data scaling issues regarding how the performance of low resource language can scale with the high resource language data size. The primary claim is that creating synthetic data by translating from high resource languages into low resource languages can not achieve cross lingual data scaling; on the other hand, pretraining models solely in high resource languages followed by continuing training with balanced mixed of high and low resource languages can help to achieve cross-language data scaling.

**Strengths:**

• The topic of scaling performance of low resource languages by scaling the data size of high resource language is attractive.
	• The epxeriments indicate that pretraining models with high resource language data followed by continuing training with balanced mixture of high- and low- resource language data is effective.

**Weaknesses:**

• High resource language pretraining followed by low resource language continuing training with carefully designed balanced data mixing and learning rates is not new.

**Questions:**

1. Line 185, a typo: a period punctuation is needed after "..for pretraining".
	2. Section 3.4, please clarify why "order adjustment" and "proportion adjustment" is deemed as knowledge-transfer? What exactly is "knowledge" here?
	3. Figure 2 (b) please label which plot is English and which is Chinese.
	4. Figure 4, how do the English tasks perform as the Chinese tasks performance increase?
	5. Line 408-413, as learning rate schedule is experimented with, how about having longer training steps with small learning rates? How to rule out the possiblity of immatured training without enough steps?

---

> ### Author Response · Authors · 2025-12-03
>
> **Q1**. High resource language pretraining followed by low resource language continuing training with carefully designed balanced data mixing and learning rates is not new.
>
> Our primary contribution is to systematically investigate a problem that has not been explored in prior work.
> To the best of our knowledge, we are the first to systematically study and successfully achieve cross-lingual data scaling, where performance in low-resource languages scales with the dataset size of high-resource languages.
>
> In this process, we employ a combination of common techniques, enabling us to reveal properties that have been largely overlooked, while incurring no extra pretraining cost.
> We consider this simplicity an advantage rather than a drawback.
>
>
> **Q2**. Section 3.4, please clarify why "order adjustment" and "proportion adjustment" is deemed as knowledge-transfer? What exactly is "knowledge" here?
>
> "Order adjustment" adjusts the training order of different language data, attempting to initialize the training of low-resource language with high-resource language data, thereby enabling knowledge transfer from high-resource language to low-resource language through initialization.
> "Proportion adjustment" is an improvement on order adjustment and is therefore also included in this section.
> Referring to the cross-lingual shared-token experiment in our paper, we tend to believe that the model learns "representation knowledge" from low-resource language tokens embedded in high-resource language data.
>
>
> **Q3**. Figure 2 (b) please label which plot is English and which is Chinese.
>
> In Figure 2, the red curve denotes the performance of Chinese, while the blue curve denotes the performance of English.
> This is reflected in the dual y-axis labels and corresponding color scheme.
> To avoid any possible ambiguity, we will add an explicit legend to the figure in the revised version.
>
> **Q4**. Figure 4, how do the English tasks perform as the Chinese tasks performance increase?
>
> In the experiments shown in Figure 4, as the amount of English data increases, the English performance consistently improves across all methods as well.
>
> | **Method**            | **200B** | **300B** | **500B** |
> |-------------------|-----------|-----------|-----------|
> | **Baseline**          | 24.520    | 27.277    | **29.487**    |
> | **Translation**       | 23.516    | 23.261    | **25.050**    |
> | **Code-switching**    | 24.920    | 27.946    | **28.782**    |
> | **Order Adjustment**  | 24.685    | 27.116    | **28.231**    |
> | **Data Repetition**   | 23.874    | 25.463    | **28.205**    |
> | **Loss Weighting**    | 23.299    | 24.076    | **26.648**    |
> | **ScaleX**            | 23.968    | 26.562    | **28.552**    |
> | **ScaleX-DR**         | 23.830    | 26.781    | **28.043**    |
>
>
> **Q5**. As learning rate schedule is experimented with, how about having longer training steps with small learning rates? How to rule out the possiblity of immatured training without enough steps?
>
> In ScaleX, when training transitions from the first stage to the second stage, the data distribution changes substantially.
> A sufficiently large learning rate is needed at this point to help the model escape the local optimum formed in the previous stage.
> If the learning rate is too small, additional training steps will only cause the model to oscillate around the existing local optimum rather than move toward a better solution.
> Moreover, increasing the number of training steps substantially increases pretraining cost, which is impractical for large-scale LLM training.
>
> In our paper, we already validated the effectiveness of ScaleX using a 2.5B LLM trained on 1T tokens.
> This large-scale experiment helps mitigate concerns about under-training and confirms that our observations are not artifacts of insufficient optimization.

---

### Meta-Review · Area_Chair_xMqV · 2026-01-07

**Summary:**

The primary concerns are regarding the novelty of the method and validity of the experimental setups.
The paper presents a systematical analysis but the techniques exist in prior works. The high resource first two-stage training process follows the standard continual training recipe. The use of Chinese as low-resource language raises a bunch of questions, whether the subsampling is representative etc. It is fine to use that for analysis but afterward further validation on real low-resource languages would make it more convincing.

**Reviewer Concerns:**

8jCx:
* novelty, two stage training follows continue training recipe: the rebuttal is not convincing
* clarity on terms like "knowledge transfer", plot labels and initial metrics are only PPL instead of downstream: addressed in the rebuttal

JH1T:
* the use of Chinese as low-resource: the authors justified based on data quality control, cited other papers doing the same, but the fundamental mismatch remains
* missing details on "ScaleX-DR": addressed in rebuttal
* translation baseline: the authors cited cost as a barrier to use closed-source high quality translation models, but that doesn't answer whether high-quality translation would bring improvements
* code-switching is done with a static ratio for simplicity, suggested to use a dynamic schedule: not addressed

d6jm:
* missing baselines: not addressed, the authors cited papers and argued they operate under different assumptions
* metrics using only PPL: addressed in rebuttal with downstream results
* concern on step-by-step approach: not addressed

77Sf:
* language choice, use Chinese as low resource: the rebuttal is not convincing
* statistical significance of the results: addressed in rebuttal

**Reviewer Scores:**

8jCx: 4, likely no change
JH1T: 6, likely no change
d6jm: 2, likely no change
77Sf: 4, likely no change

---

### Decision · Program_Chairs · 2026-01-26

Reject